

# Full-physics carbon dioxide retrievals from the OCO-2 satellite by only using the 2.06 $\mu$m band

Lianghai Wu[1], Otto Hasekamp[1], Haili Hu[1], Joost aan de Brugh[1], Jochen Landgraf[1], Andre Butz[2], and Ilse Aben[1]

[1]SRON Netherlands Institute for Space Research, Utrecht, The Netherlands
[2]Institute of Environmental Physics, University of Heidelberg, Heidelberg, Germany

**Correspondence:** Lianghai Wu (L.Wu@sron.nl)

**Abstract.** Passive remote sensing of atmospheric carbon dioxide uses spectroscopic measurements of sunlight back-scattered by the Earth's surface and atmosphere. The current state-of-the-art retrieval methods use three different spectral bands, the oxygen A band at $0.76$ $\mu$m and the weak and strong $CO_2$ absorption bands at $1.61$ and $2.06$ $\mu$m, respectively, to infer information on light scattering and the carbon dioxide column-averaged dry-air mole fraction $XCO_2$. In this study, we propose a
one-band $XCO_2$ retrieval technique which uses only the $2.06$ $\mu$m band measurements from the OCO-2 satellite. We examine the data quality by comparing the OCO-2 $XCO_2$ with collocated ground based measurements from the Total Carbon Column Observing Network (TCCON). Over land and ocean the OCO-2 one-band retrieval shows differences to TCCON observations with a standard deviation of $\sim 1.30$ ppm and a station-to-station variability of $\sim 0.50$ ppm. Moreover, we compare one-band and three-band retrievals over Europe,the Middle East and Africa region and see high correlation between the two retrievals
with a SD of 0.93 ppm. Compared to the three-band retrievals, using only the $2.06$ $\mu$m band similar $XCO_2$ retrieval accuracy and precision can be obtained while retaining a similar data yield.

## 1 Introduction

Since the past decade, space-based measurements of atmospheric carbon dioxide ($CO_2$) are used, along with ground-based measurements, to characterize $CO_2$ sources and sinks in order to better understand the carbon cycle. The inversion models that calculate the $CO_2$ fluxes are sensitive to biases in the carbon dioxide dry-air column-averaged mole fraction ($XCO_2$)
as small as $0.5$ ppm (see e.g. Miller et al. (2007); Basu et al. (2013)). This poses enormous challenges on the instruments, calibration and retrieval algorithms used to measure $XCO_2$ and much effort is needed to reduce e.g. instrument, calibration, spectroscopy and other forward model errors. In particular, scattering by aerosol and thin cirrus clouds (thick clouds are screened) can lead to light path modifications causing unacceptable errors in $XCO_2$ if not accounted for in the radiative
transfer calculations (Guerlet et al., 2013; Aben et al., 2007). The currently operational $CO_2$ satellites, i.e. the Greenhouse gases Observing Satellite (GOSAT) (Kuze et al., 2009) and the Orbiting Carbon Observatory-2 (OCO-2) (Crisp et al., 2017), and the corresponding retrieval algorithms (e.g. Butz et al. (2009); Boesch et al. (2011); O'Dell et al. (2012); Buchwitz et al. (2017)) apply a three-band approach using three spectral bands around 0.76 ($O_2$ A-band), 1.61 (weak $CO_2$ band) and 2.06



$\mu$m (strong $CO_2$ band) to simultaneously retrieve $XCO_2$ and other relevant parameters such as surface albedos and aerosol properties.

It has been proposed by Butz et al. (2009), based on simulated OCO measurements, that retrievals using the 2.06 $\mu$m band alone actually show similar performance as using three bands. The reasons would be that one-band retrievals are less dependent

on spectral scattering properties than three-band retrievals. We examine whether this claim holds for real OCO-2 measurements, by comparing the $XCO_2$ products for both methods in terms of accuracy and data yield. For the OCO-2 measurements a single band retrieval is computationally less expensive, which is important considering the huge data amount to be processed. More generally, a single-band retrieval requires a more simple and thus cheaper instrument, and may avoid possible complications related to spectral window-dependent (calibration) errors. For example for OCO-2, there are indications that it is necessary

to fit an intensity offset in the weak and strong $CO_2$ absorption bands to account for potential instrumental errors (Wu et al., 2018).

The paper is organized as follows: we first introduce the data that we used in this work in Sect. 2. The retrieval algorithm and setup for three-band and one-band retrievals are described in Sect. 3. Section 4 evaluates the one-band retrieval performance with TCCON $XCO_2$ observations and compares the performance to that of three-band retrievals. Finally we conclude and

discuss our findings in Sect. 5.

## 2   Data

In this paper, we use OCO-2 version 8 L1b data between September 2014 and October 2017. To evaluate the retrieval performance, we only use measurements that are collocated with TCCON measurements. Although some limitations exist as discussed by Kulawik et al. (2016), TCCON measurements are still the most appropriate validation product for space based

$XCO_2$ retrievals. OCO-2 measurements are considered collocated when they are taken within 2 hours and a distance of less than 3 degrees in both latitude and longitude of a TCCON measurement. Here, we do not use TCCON stations located within polluted areas, high latitude regions or areas with significant topography. The retrieval algorithm also uses the ECMWF (European Centre for Medium Range Weather Forecasts) high-resolution analysis data to get meteorological information including pressure, temperature, humidity and surface wind speed. For each OCO-2 measurement, the surface elevation data is obtained

from the 90 m digital elevation data of NASA's Shuttle Radar Topography Mission (SRTM) (Farr et al., 2007). Prior information on the carbon dioxide profile is extracted from the CarbonTracker model for the year 2013 with an added annual increase of 2.25 ppm (Peters et al., 2007).

## 3   Retrieval algorithm and methodology

We use the RemoTeC retrieval algorithm (Hasekamp and Butz, 2008; Butz et al., 2009), which has been extensively used for

greenhouse gas retrievals from satellite observations like GOSAT, OCO-2 and S5P measurements (Butz et al., 2011; Schepers et al., 2012; Guerlet et al., 2013; Hu et al., 2016; Wu et al., 2018; Hu et al., 2018). The adaptations and first use for OCO-2





measurements are described in Wu et al. (2018). There we employed the three-band $XCO_2$ retrievals which will be used here as a reference to compare against our $XCO_2$ retrievals from the 2.06 $\mu$m band.

The three-band retrieval fits OCO-2 measurements in all the three OCO-2 spectral windows. The state vector that is retrieved contains 35 elements as shown in Table 1 : a 12-layer vertical profile of $CO_2$ partial columns, the total columns of $H_2O$ and

$CH_4$, three effective scattering parameters and, for each channel, three albedo parameters describing the Lambertian albedo up to its second order spectral dependence, an intensity offset, spectral shifts for the Earth radiance measurement and the solar reference model. We do not retrieve the dry air column but compute it using the ECMWF meteorological data. As described in Wu et al. (2018), we use a Lambertian reflection model for land surface reflection properties and for ocean surfaces a wind-speed driven reflection model of Cox and Munk (1954) combined with an additive wavelength-dependent Lambertian term.

The retrieved three aerosol parameters are the total column number density $N$, the parameter $\alpha$ of a power-law size distribution ($n(r) \propto r^{-\alpha}$ with the particle radius $r$) and the central height parameter $z$ of a Gaussian height distribution. The full width half maximum of the Gaussian height distribution is fixed at 2 km.

In the one-band retrieval, we attempt to infer $XCO_2$ by only using OCO-2 measurements in the spectral range 2042-2081 nm. The state vector is the same as for the three-band retrieval except that the $CH_4$ column is not included and it only contains

surface albedo, intensity offset and spectral shift parameters for the 2.06 $\mu$m band (see Table 1). In the retrieval, we seek the state vector for which a cost function including the difference between the forward model and measurements and a side constraint is minimized. The same Phillips-Tikhonov regularization scheme as employed in the three-band retrieval is used to solve the minimization problem iteratively (Phillips, 1962; Tikhonov, 1963; Hasekamp and Landgraf, 2005; Wu et al., 2018). Like for the three-band retrieval, we choose the regularization parameter such that the degree of freedom for signal (DFS) for

the $CO_2$ profile is in the range 1.0-1.5 (Wu et al., 2018).

It should be noted that the retrieval algorithm is only applicable to clear-sky scenes, so we must define a suitable cloud filter that preselects the scenes to be processed. Before performing full-physics $XCO_2$ retrievals, we retrieve the columns of $O_2$, $CO_2$ and $H_2O$ independently in the three spectral bands under the assumption of a non-scattering atmosphere. When neglecting cloud or aerosol scattering, the ratio between the $CO_2$ or $H_2O$ column retrieved from the 1.61 $\mu$m band and those

retrieved from the 2.06 $\mu$m band is a measure of the lightpath modification because a large deviation can be introduced due to different light path sensitivity. The ratio between the retrieved $O_2$ column and the one computed from the ECMWF surface pressure can also be used to detect clouds. We consider the following scenes as sufficiently cloud-free:

$$0.90 < \frac{O_2(0.76\mu m)}{O_2(ecmwf)} < 1.02, 0.98 < \frac{H_2O(1.61\mu m)}{H_2O(2.06\mu m)} < 1.05 \text{ and } 0.98 < \frac{CO_2(1.61\mu m)}{CO_2(2.06\mu m)} < 1.03 \qquad (1)$$

This classifies around 26% of all soundings as cloud-free. However, this cloud screening strategy can not work for the

one-band retrieval because here we restrict ourselves to use only measurements from the 2.06 $\mu$m band.

Here, we propose a new cloudfilter based only on the 2.06 $\mu$m band, to truly investigate the case where the other bands are not available. We first screen by retrieving $XCO_2$ using the whole 2.06 $\mu$m band under the assumption of a non-scattering atmosphere and divide this by the a priori value derived from the CarbonTracker. When this ratio is $< 0.96$ or $> 1.04$, the scene is considered too cloudy for $XCO_2$ retrieval. Then, we use two sub spectral windows in the 2.06 $\mu$m band: one weak





absorption window centered around 2.08 $\mu$m in the spectral range 2078-2081 nm and one strong absorption window centered around 2.05 $\mu$m in the spectral range 2042-2057 nm. The columns of $CO_2$ and $H_2O$ are retrieved independently from these two sub windows under the assumption of a non-scattering atmosphere, and the ratios between the $CO_2$ or $H_2O$ column retrieved from those two sub windows are used for cloud screening. We only use spectra which meet the following criteria:

$$5 \quad 0.89 < \frac{H_2O(2.08\mu m)}{H_2O(2.05\mu m)} < 1.05 \text{ and } 0.98 < \frac{CO_2(2.08\mu m)}{CO_2(2.05\mu m)} < 1.03 \tag{2}$$

After the filtering procedure described above, around 27% of total soundings are considered as cloud-free cases, which is similar to what is found by the three-band cloudfilter. The one-band cloudfilter and the three-band cloudfilter have an overlap of 75%.

For cloud-screened soundings, we first run full-physics retrievals and then apply posterior quality filtering based on the
criteria shown in Table 2. Those criteria are related to extreme viewing geometry, difficult scattering scenes, challenging surface properties, spectra with larger uncertainties and poor fit between forward model and measurements. After the quality filtering, the overall throughputs are 17.0% and 18.0% for one-band and three-band retrievals, respectively. The two data sets have an overlap of 75%.

## 4 Performance evaluation

Note that in this work, we do not apply a bias correction as it is common practice for $CO_2$ retrievals from space based observations (Wunch et al., 2017; Wu et al., 2018; O'Dell et al., 2018), but show the uncorrected results because we want to evaluate the true retrieval capability. Due to the high spatial sampling of OCO-2, we typically obtain several collocations of OCO-2 retrievals with individual TCCON measurements for our collocation criteria in a single overpass. To reduce the impact of random and representation errors in our comparison, we compare overpass-averages between OCO-2 and TCCON results and use
bias ($b_a$), standard deviation of the difference ($\sigma_a$) and station-to-station variability ($\sigma_s$) for performance evaluation (Buchwitz et al., 2017). The station-to-station variability is the standard deviation of all biases between the different TCCON sites and is a measure of regional-scale accuracy which is crucial for flux inversion.

Figure 1 shows validations of one-band $XCO_2$ retrievals over land and ocean. We neglect cases where less than 10 individual data points are available in OCO-2 retrievals during one overpass. Here, both land and ocean retrievals exhibit high correlation
(around 0.94) with TCCON data and both have an standard deviation (SD) of $\sim$ 1.30 ppm.

To evaluate the one-band and three-band retrieval performance in more detail, Figs. 2 and 3 show the bias and SD of the retrievals per TCCON station. One-band and three-band retrievals have similar bias and SD among most individual stations. One-band retrievals have slightly higher overall SD which is increased by 0.01 ppm for land retrievals and 0.14 ppm for ocean retrievals. Over land, one-band and three-band retrievals have comparable station-to-station variability of 0.44 and 0.42 ppm,
respectively. Over ocean, the one-band retrieval has a station-to-station variability of 0.55 ppm which is about 0.1 ppm higher than that of three-band retrieval, however, as shown in Fig. 3 this is mainly caused by larger biases from the Lauder and Ascension stations.



Table 3 summarizes the overall validation performance of the one-band and three-band retrievals with TCCON measurements. Compared with three-band retrievals, one-band retrievals have similar throughput and similar high correlation coefficients with TCCON. In one-band retrievals, the single sounding precision are 0.16 ppm larger over both land and ocean. In term of bias, one-band retrievals have a smaller overall bias but a similar station-to-station variability as three-band retrievals.

For the general applicability of the one-band retrieval, it is important to know if the performance of the one-band retrievals is more affected by the amount and properties of aerosols than the three-band retrievals. Figure 4 shows one-band and three-band land retrieval differences with respect to TCCON as a function of aerosol optical thickness (AOT) in the $O_2$ A band, size parameter and layer height as retrieved by the three-band retrievals. With AOT, one-band retrievals show a positive correlation of 0.17 while three-band retrievals present a anti-correlation of $-0.11$. In both retrievals, the range of errors between AOT$=$

0.01 and AOT$= 0.30$ are around 1.0 ppm. Scattering errors in both retrievals shows similar correlations with aerosol size parameter and layer central height. So, compared with three-band retrievals, one-band retrievals exhibit a similar dependence on aerosol properties.

On the other hand, the DFS for aerosol parameters in the one-band retrieval is mostly well below 1 while for the three-band retrieval it is around 2 in most cases. This triggers the question whether a non-scattering retrieval would also provide

similar performance as the one-band "full-physics" retrieval for the cases considered in this study. To investigate this, we also performed a non-scattering retrieval using the 2.06 $\mu$m band only. The results are summarized in Table 4. It can be seen that the non-scattering retrieval have a much larger bias, and the standard deviation of differences with TCCON and the station-to-station bias are somewhat larger than for the one-band (and three-band) retrieval. The improvement of the one-band retrievals compared to the non-scattering retrievals becomes more clear if we consider the Izana TCCON station close to the Sahara,

known as a region with difficult aerosol scenes for $XCO_2$ retrieval. Here, we employ a coarse spatial collocation criteria (16.5 $<$ latitude $<$ 34.0 degrees and $-16.0 <$ longitude $<$ 24.5 degrees) for observations made between September 2014 and October 2017 which results in 100 thousands valid retrievals. For this Sahara region, the standard deviation of differences with TCCON for one-band ad non-scattering retrievals are 1.49 and 1.93 ppm, respectively.

We conclude that despite the small DFS for aerosol properties in the one-band retrieval, the explicit treatment of aerosols in

the one-band retrieval is still important to achieve sufficient accuracy on $XCO_2$, comparable to the three-band retrievals.

To further investigate the validity of the conclusions based on the OCO-2 vs TCCON comparison, we performed a comparison between one-band and three-band retrievals over a larger region. Here, we do one-band and three-band $XCO_2$ retrievals over Europe, the Middle East and Africa (EMEA) region for all OCO-2 observations made between 08 September 2014 and 31 December 2014. In Fig. 5, one-band and three-band retrievals over the EMEA region show similar data coverage and regional

$XCO_2$ variations. For example, low $XCO_2$ values in East Europe and enhancement towards the Middle East. Here, one-band and three-band retrievals are highly correlated ($r = 0.84$) with a SD of 0.93 ppm. This indicates that the conclusions drawn above on the similar performance between the one-band and three-band retrievals are not only valid for regions around TCCON stations.





# 5 Conclusions

The comparison between the performance of one-band $XCO_2$ retrievals from OCO-2 using only the 2.06 $\mu$m band and the commonly employed three-band retrievals showed that with one band similar accuracy can be achieved as with three bands while the processing time is reduced by 40%. The most noticeable difference is the slightly increased standard deviation of the

differences between OCO-2 and TCCON measurements. We see that leaving out the $O_2$ A-band and weak $CO_2$ absorption band has little effect on the station-to-station variability in the $XCO_2$ retrievals. Our results suggest that the $O_2$ A-band adds only limited information on aerosols relevant for $XCO_2$ retrievals confirming earlier results (Butz et al., 2009) using simulated OCO measurements. For future missions it may be better to replace the $O_2$ A-band with measurements that have larger information content on aerosols, like a Multi-Angle Polarimeter (MAP) (Mishchenko and Travis, 1997; Hasekamp and Landgraf, 2007;

Wu et al., 2015).

In order to evaluate the true retrieval capability of the the one-band and three-band retrievals, we have not applied any bias corrections in this study. It should be noted though, that in general a bias correction is needed and will improve the validation against TCCON. For example, Wunch et al. (2017) and Kiel et al. (2018) have found it necessary to apply, among other, a swath-dependent bias correction.

*Data availability.* The OCO-2 L1b data (version 8) were provided by the OCO-2 project from the data archive at the NASA Goddard Earth Science Data and Information Services Center (https://daac.gsfc.nasa.gov/). TCCON data were obtained from the TCCON Data Archive (https://tccon-wiki.caltech.edu/). The three-band and one-band retrieval results presented in this paper can be found at ftp://ftp.sron.nl/open-access-data/

*Competing interests.* The authors declare that there is no conflict of interest.

*Acknowledgements.* This research was funded by the Netherlands Space Office as part of the User Support Programme Space Research under project ALW-GO/15-23. We would like to thank TCCON community for providing measurements used in this study.



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





**Table 1.** State vector elements for the three-band and one-band retrievals.

| state vector elements | three-band | one-band | A priori in one-band retrieval |
|---|---|---|---|
| $CO_2$ sub-columns in 12 vertical layers | 12 | 12 | CarbonTracker 2013 |
| $CH_4$ total column | 1 | 0 | - |
| $H_2O$ total column | 1 | 1 | ECMWF |
| aerosol column $N$ | 1 | 1 | $2.18 \times 10^{11}$ m$^{-2}$ ($\tau = 0.02$ in 2.06 $\mu$m band) |
| aerosol size parameter $\alpha$ | 1 | 1 | 4.0 |
| aerosol height parameter $z$ | 1 | 1 | 2000 m |
| albedo parameters | 9 | 3 | Estimated from measured radiance |
| spectral shift Earth radiance spectrum | 3 | 1 | 0.0 |
| spectral shift solar reference spectrum | 3 | 1 | 0.0 |
| intensity offset | 3 | 1 | 0.0 |

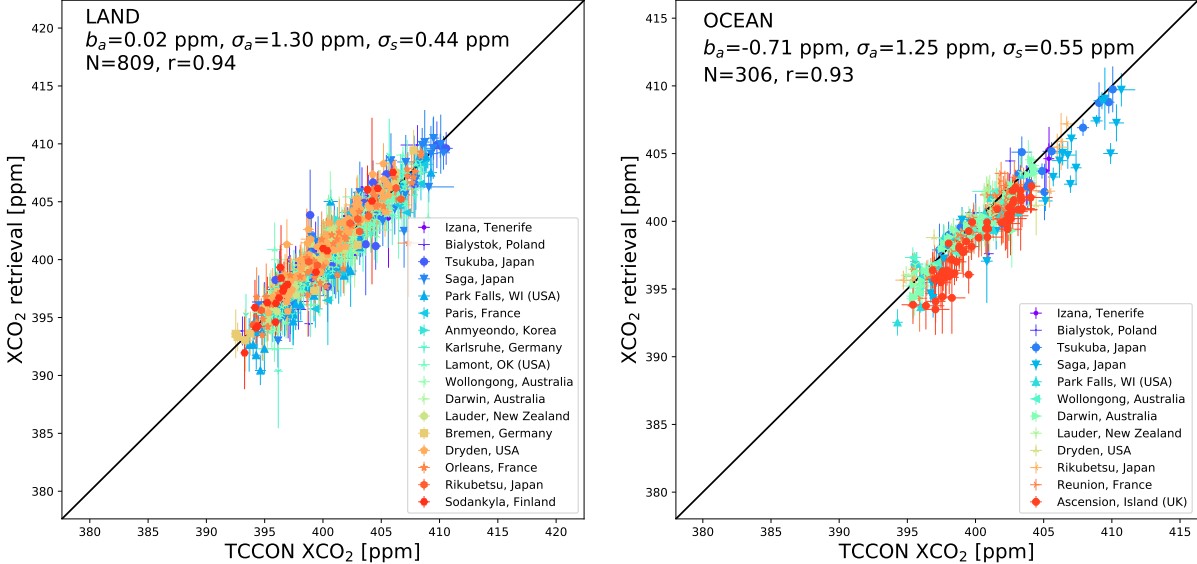

**Figure 1.** XCO$_2$ retrievals by using only the 2.06 $\mu$m band of OCO-2. We evaluate overpass averaged results over land and ocean separately. In each panel, we include bias ($b_a$), standard deviation of the difference ($\sigma_a$), station-to-station variability ($\sigma_s$), number of overpass (N), Pearson correlation coefficient ($r$) and one-to-one line. For each overpass, variations of XCO$_2$ retrievals and TCCON data are presented with errorbars.

**Table 2.** Filter variables applied to reject low quality $XCO_2$ retrievals over land and ocean in three-band and one-band retrievals. For most variables, ocean glint retrievals have the same filtering criteria as those over land. However, due to ocean-glint's unique viewing geometry and different surface properties, aerosol and surface related filtering variables have different ranges and are listed separately in brackets. Filter variables not used in the relevant retrieval type are marked with a $\forall$ sign. The blended albedo can be derived using surface albedos in $O_2$ A-band ($A_{0.76}$) and 2.06 $\mu$m band ($A_{2.06}$) by $2.4A_{0.76}-1.13A_{2.06}$ (Wunch et al., 2011). The aerosol ratio parameter is calculated with the three retrieved aerosol parameters by $\tau * z/\alpha$.

| Filter variables | three-band retrieval | one-band retrieval |
|---|---|---|
| Solar zenith angle | $\leq$ 75 degrees | $\leq$ 75 degrees |
| Viewing zenith angle | $\leq$ 45 degrees | $\leq$ 45 degrees |
| Surface elevation variation | $\leq$ 75 meters | $\leq$ 75 meters |
| Degrees of freedom for signal for $CO_2$ | > 1.0 | > 1.0 |
| Signal-to-noise ratio (SNR) in $O_2$ A-band | >= 100.0 | $\forall$ |
| SNR in 2.06 $\mu$m band | >= 100.0 | >= 100.0 |
| Overall goodness of fit | <= 35.0 | $\forall$ |
| Goodness of fit in $O_2$ A-band | <= 35.0 | $\forall$ |
| Goodness of fit in 2.06 $\mu$m band | $\forall$ | <= 35.0 |
| Blended albedo | <= 1.0 | $\forall$ |
| Albedo slope in 2.06 $\mu$m band | $\forall$ | $-0.0001$ <=and<= 0.0005(0.00004 <=and<= 0.0003) |
| Aerosol size parameter | 3.0 <=and<= 8.5(3.0 <=and<= 5.0) | 3.5 <=and<= 5.0(3.995 <=and<= 4.05) |
| Aerosol optical depth in $O_2$ A-band | <= 0.35(<= 0.55) | $\forall$ |
| Aerosol optical depth in 2.06 $\mu$m band | $\forall$ | <= 0.1 |
| Aerosol ratio parameter | <= 300 | <= 300 |
| Ratio of $CO_2$ between non-scattering and full-physics retrievals | $\forall$ | 0.985 <=and<= 1.01 |
| Ratio of $H_2O$ between non-scattering and full-physics retrievals | $\forall$ | 0.975 <=and<= 1.01 |
| Retrieval uncertainty for $XCO_2$ | <= 1.0 | <= 1.0 |
| Fitted intensity offset ratio in $O_2$ A-band | $-0.005$ <=and<= 0.015 | $\forall$ |
| Fitted intensity offset ratio in 1.61 $\mu$m band | $-0.005$ <=and<= 0.015 | $\forall$ |
| Fitted intensity offset ratio in 2.06 $\mu$m band | $-0.005$ <=and>= 0.015 | $-0.005$ <=and<= 0.015 |
| Added Lambertian term in 2.06 $\mu$m band | $\forall$(<= 0.65) | $\forall$(<= 0.65) |





**Table 3.** Overall performance of three-band and one-band retrievals. Here, overall bias and single sounding precision are estimated for single soundings. All other quantities are obtained using overpass averaged values.

| Diagnostics | three-band | | One-band | |
|---|---|---|---|---|
| | land | ocean | land | ocean |
| number of valid retrievals [thousand] | 366.5 | 135.6 | 343.2 | 130.3 |
| overall bias $b$ [ppm] | 0.88 | 1.54 | −0.12 | −0.76 |
| single sounding precision $\sigma$ [ppm] | 1.65 | 1.59 | 1.81 | 1.75 |
| number of overpass | 816 | 300 | 809 | 306 |
| bias $b_a$ [ppm] | 1.05 | 1.42 | 0.02 | −0.71 |
| standard deviation (SD) $\sigma_a$ [ppm] | 1.29 | 1.11 | 1.30 | 1.25 |
| station-to-station variability $\sigma_s$ [ppm] | 0.42 | 0.46 | 0.44 | 0.55 |
| Pearson correlation coefficient(cor) | 0.94 | 0.94 | 0.94 | 0.93 |
| mean CPU time per retrieval | 21.0 seconds | | 13.0 seconds | |

**Table 4.** Similar as Table 3 but for non-scattering retrievals using the 2.06 $\mu$m band. Here, we use the same cases as one-band retrievals in Table 3

| Diagnostics | Non-scattering | |
|---|---|---|
| | land | ocean |
| overall bias $b$ [ppm] | −4.27 | −5.15 |
| single sounding precision $\sigma$ [ppm] | 1.87 | 1.85 |
| bias $b_a$ [ppm] | −4.36 | −5.31 |
| standard deviation (SD) $\sigma_a$ [ppm] | 1.34 | 1.40 |
| station-to-station variability $\sigma_s$ [ppm] | 0.53 | 0.59 |
| Pearson correlation coefficient(cor) | 0.93 | 0.91 |



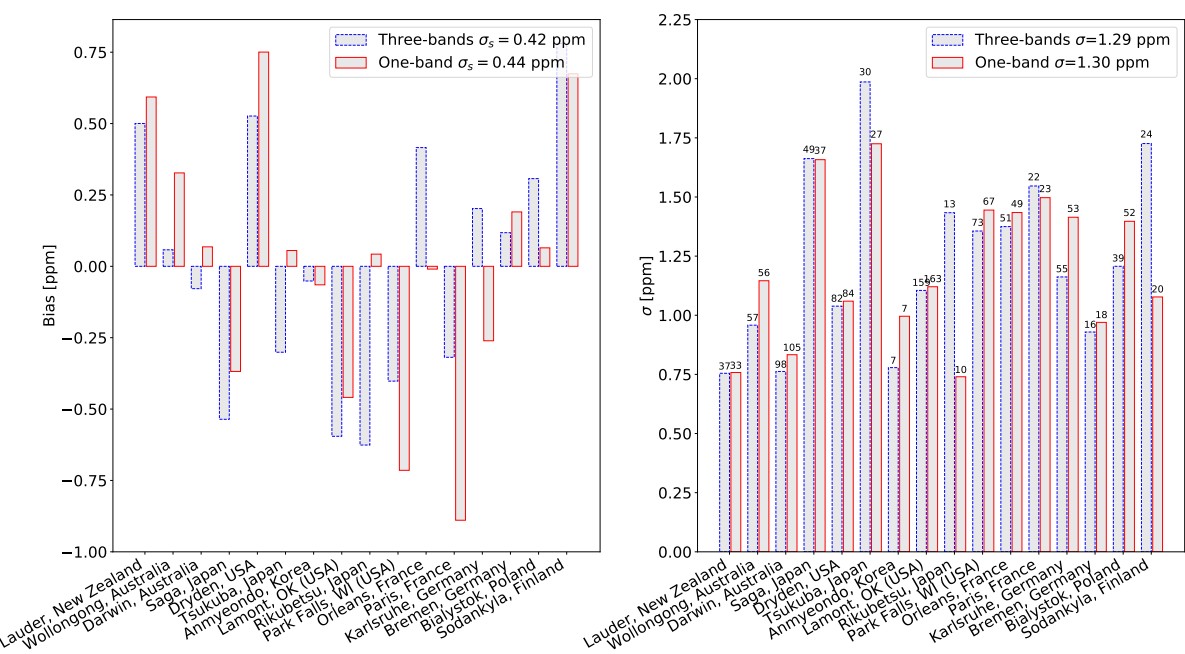

**Figure 2.** Bias (left panel) and standard deviation (right panel) variation at different TCCON stations for one-band and three-band retrievals over land. To see the bias variation on the same reference level, we directly subtract mean bias $b_a$ of one-band and three-band retrievals accordingly as listed in Table 3. The station-to-station variability ($\sigma_s$) is included in the legend of left panel. In the right panel, number of overpass at each station is listed on the bar. The TCCON stations are ordered by latitude from southern hemisphere to northern hemisphere. Stations with less than 5 overpasses are excluded.





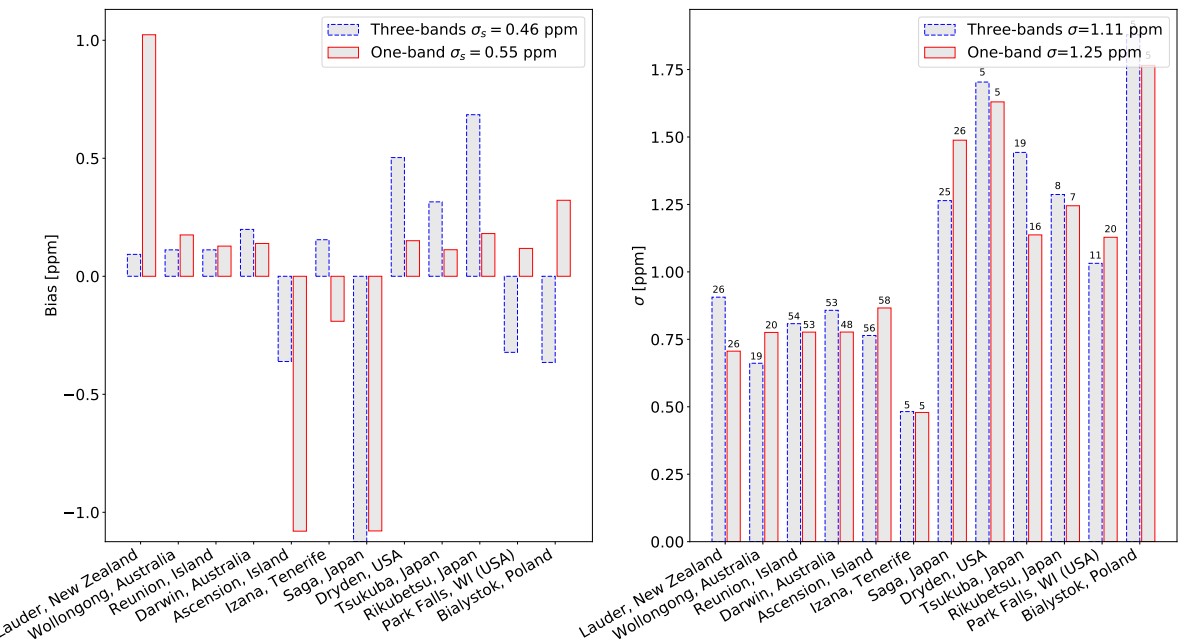

**Figure 3.** Same as Fig. 2, but for retrievals over ocean.

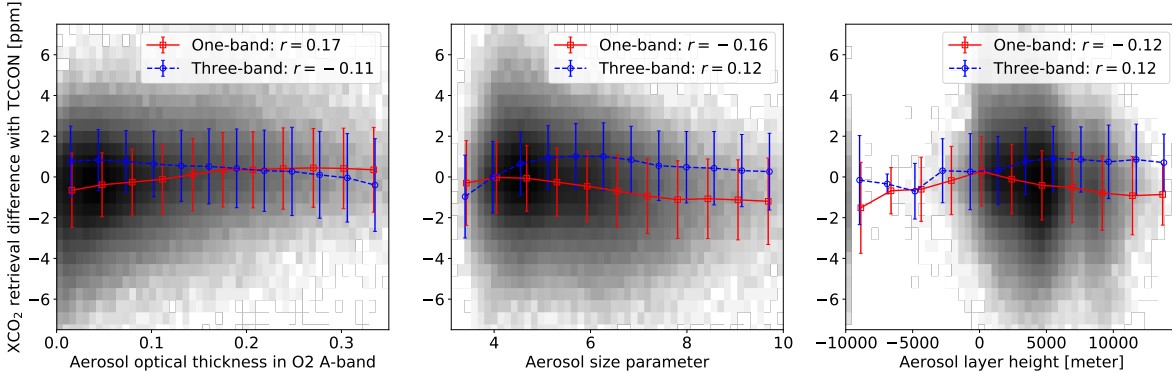

**Figure 4.** Error on $XCO_2$ from one-band and three-band OCO-2 land retrievals as a function of aerosol optical thickness (in $O_2$ A-band), size parameter and layer height as retrieved by three-band retrievals. Shown are the mean bias for each parameter bin along with standard deviation within each bin. Background includes the density map of $XCO_2$ errors from one-band retrievals.

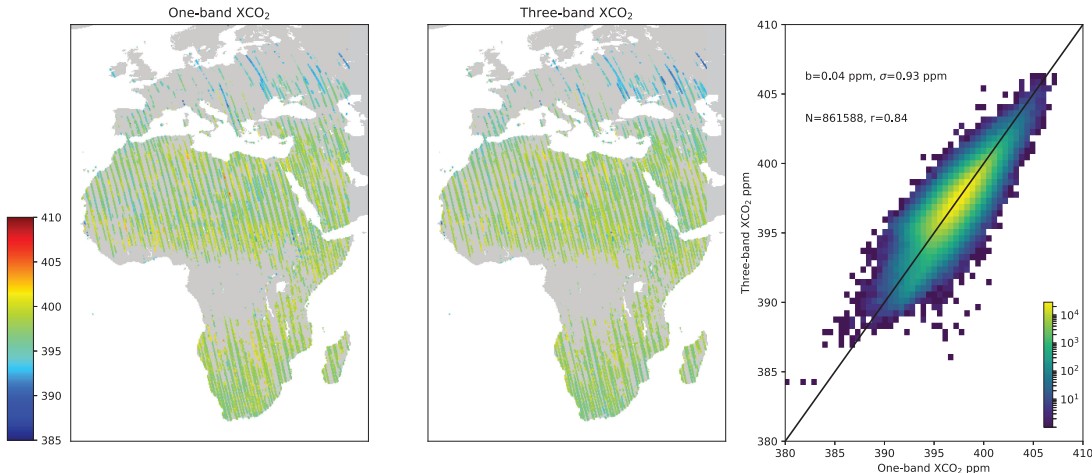

**Figure 5.** $XCO_2$ distributions over the EMEA regions from one-band and three-band retrievals in the time period between 08 September 2014 and 31 December 2014. In the most right panel, corresponding $XCO_2$ retrievals from one-band and three-band retrievals are shown with bias($b$), standard deviation ($\sigma$) and Pearson correlation coefficient ($r$). Here, a mean bias of $0.88$ ppm was subtracted from three-band retrievals.