# Peer review of "Full-physics carbon dioxide retrievals from the OCO-2 satellite by only using the 2.06 $\mu$ m band"

_Atmospheric Measurement Techniques, 2019_

## Referee Comment (RC1) · Anonymous Referee #1 · 11 Jul 2019

Interactive comment on the manuscript "Full-physics carbon dioxide retrievals from the OCO-2 satellite by only using the 2.06 $\mu$m band" by Lianghai Wu et al. The manuscript "Full-physics carbon dioxide retrievals from the OCO-2 satellite by only using the 2.06 $\mu$m band" contains important new material and it covers the topics appropriate for Atmos. Meas. Tech. The presented results are of practical interest in terms of reducing computational costs as well as optimizing the configuration of the measuring tools for monitoring atmospheric carbon dioxide. The manuscript is well structured and written. The abstract clearly summarizes the paper and main results. I recommend the manuscript publication provided some minor comments would be considered. 1) The proposed algorithm modification (reduction of the input spectroscopic data from three bands to one 2.06 $\mu$m- band) has been implemented for RemoteC algorithm. Specific

feature of this algorithm is using a priori (meteorological) surface pressure. The authors mentioned it ("We do not retrieve the dry air column but compute it using the ECMWF meteorological data", page 3, line 8). To my opinion, the importance of this feature for the implementation of 1-band version should be clearly noted in the discussion. In the similar algorithms (e. g., ACOS, NIES-GOSAT, and TANSAT) that retrieve surface pressure, the excluding the oxygen A band from the input spectroscopic data is hardly possible.

2) The modified (1-band) algorithm is supplemented by new cloud filtering procedure. The algorithm itself was previously tested on simulated OCO measurements. Has the filtering procedure been tested in the similar way?

3) As follows from table 4, light-scattering by aerosols for the collocated OCO-TCCON observations mostly reduces optical path-length both over ocean (quite predictable), and over land. This reduction is rather successfully corrected by 1-band algorithm in terms of $XCO_2$ biases (table 3). To demonstrate algorithm accuracy under different aerosol conditions, it would be useful to show the $XCO_2$ biases (in addition to SD values) for the Sahara region, where we can expect an increase in optical path-length by light-scattering.

---

## Referee Comment (RC2) · Robert Roland Nelson (Referee) · 23 Jul 2019

General comments:

The manuscript entitled, "Full-physics carbon dioxide retrievals from the OCO-2 satellite by only using the 2.06 $\mu$m band" presents a discussion on a non-traditional method of retrieving the column-averaged dry-air mole fraction of carbon dioxide (XCO2) from OCO-2. The authors extend the simulated work of Butz et al. (2009) to real OCO-2 measurements with the goal of improving the precision and accuracy of OCO-2 measurements and thus it is scientifically relevant, as current XCO2 retrieval biases are likely still too large to satisfy the demands of the carbon flux model community. Additionally, the implication that perhaps only a single-band instrument may be needed

to make high-quality XCO2 measurements from space has significant implications for potential future GHG missions. The manuscript, although brief, is presented well and I recommend publication in AMT after the authors address a few minor and technical issues.

Specific comments:

- Regarding not using a bias correction, you mention wanting to evaluate the "true" retrieval capability but, as you stated, a bias correction is always employed operationally. Did you look at implementing a bias correction for the one-band retrieval and how it impacted the final $\sigma$ values relative to the three-band retrieval?

- What do the aerosol results (AOD, size parameter, height) look like for the one-band retrieval? Are they similar to the three-band retrieval or does the lack of spectral information at 0.76 and 1.61 um cause the one-band retrieval to behave in interesting ways? In Butz et al. (2009) the size parameter is not retrieved so it would be informative to see the DFS for the three aerosol parameters retrieved in your one-band setup. In the end, it's only the XCO2 that matters but this is an important topic that at least deserves a discussion.

- The only spatial results shown are limited to four fall/winter months in 2014 over EMEA (Fig. 5). However, multiple studies have highlighted temporal patterns in OCO-2 errors (e.g. O'Dell et al., 2018). Did you look at other regions (could you show a global map?) and would it be possible to examine at least one full year of data to ensure that the one-band retrieval has no significant seasonal/regional biases relative to the three-band retrieval? Examining more regions with better coverage could reveal places where the one-band retrieval performs better or worse than the three-band, e.g. snow/ice or tropical forests.

- P3 L33: What percent of soundings are removed by comparing the non-scattering 2.06 um CO2 retrieval to CarbonTracker and filtering the ratio between 0.96 and 1.04? And how were the 0.96 and 1.04 thresholds determined? While this range is several

ppm of XCO2, potentially real signals (e.g. large power plants) might be filtered out.

- P3 L34: Could you include a physical explanation of how pre-filtering on CO2 and H2O ratios derived at 2.08 and 2.05 um works?

- Regarding Fig. 3, do you have a hypothesis as to why the one-band retrieval does poorly over Lauder and Ascension?

Technical comments:

Overall: define acronyms and technical terms before use. E.g. OCO-2, SD, "full-physics", DFS

P1 L3: change to "A-band", and on P5 L7

P1 L6: change to "ground-based"

P1 L9: remove "region"

P1 L10: Last sentence doesn't make sense

P1 L13: change to "Over the past decade"

P2 L8: change to "simpler"

P4 L7: change to "cloud filter"

P5 L17: change to "has a much"

P5 L22: change to "hundreds of thousands"

P5 L25: "and"

Figure 4: third panel, what do the aerosol layers at approximately -10000 meters represent?

Figure 5: I would recommend using a perceptually uniform colormap for plotting XCO2 (like you did for the third panel in this figure) and reducing the range so that differences

are more visible. Please put a label and units on the colorbars as well.

---

## Author Comment (AC1) · 10 Sep 2019

1) The proposed algorithm modification (reduction of the input spectroscopic data from three bands to one 2.06 µm- band) has been implemented for RemoteC algorithm. Specific feature of this algorithm is using a priori (meteorological) surface pressure. The authors mentioned it ("We do not retrieve the dry air column but compute it using the ECMWF meteorological data", page 3, line 8). To my opinion, the importance of this feature for the implementation of 1-band version should be clearly noted in the discussion. In the similar algorithms (e. g., ACOS, NIES-GOSAT, and TANSAT) that retrieve surface pressure, the excluding the oxygen A band from the input spectroscopic data is hardly possible.

R1-Indeed, many $XCO_2$ retrieval algorithms need the O2-A band to retrieve surface pressure to derive XCO2 values. This has to be mentioned clearly. A new sentence "Clearly, a one band retrieval using only the 2.06 micron band is only possible if surface pressure information from meteorological re-analysis/ forecast is used in the retrieval algorithm. Retrieving this information, as is done by most algorithms (list references here rather than algorithm names) requires the O2 a-band. ." is now added in Page 3, line 20.

2) The modified (1-band) algorithm is supplemented by new cloud filtering procedure. The algorithm itself was previously tested on simulated OCO measurements. Has the filtering procedure been tested in the similar way?

R2- The new cloud filtering has not been tested with synthetic data yet. For three-band cloud filtering, we do non-scattering retrievals in weak and strong absorption band and the idea is that the difference is a measure for light path modifications (e.g. by clouds), as weak and strong absorption bands have different light path sensitivity. The new one-band cloud filtering is based on a similar idea as the three-band cloud filtering, namely dividing the 2.06 um band into one weak absorption band around 2.08 um and one strong absorption band around 2.05 um. With real OCO-2 data used in the paper, we see that one-band cloud filtering and three-band cloud filterings have similar overall throughputs with an overlap of 75%, as stated in the manuscript

3) As follows from table 4, light-scattering by aerosols for the collocated OCO-TCCON observations mostly reduces optical path-length both over ocean (quite predictable), and over land. This reduction is rather successfully corrected by 1-band algorithm in terms of XCO2 biases (table 3). To demonstrate algorithm accuracy under different aerosol conditions, it would be useful to show the XCO2 biases (in addition to SD values) for the Sahara region, where we can expect an increase in optical path-length by light-scattering.

R3- Indeed, it is useful to mention the biases as well. For the Sahara region, biases with TCCON for one-band and non-scattering retrievals are -0.23 and -2.46 ppm, respectively. The biases for the Sahara region are now included in the paper in page 5 line 23.

---

## Author Comment (AC2) · 10 Sep 2019

We thank all reviewers for their constructive comments, which helped to improve the paper. Below, we address all comments point-by-point.

Reviewer 2#

The manuscript entitled, "Full-physics carbon dioxide retrievals from the OCO-2 satellite by only using the 2.06 µm band" presents a discussion on a non-traditional method of retrieving the column-averaged dry-air mole fraction of carbon dioxide (XCO2) from OCO-2. The authors extend the simulated work of Butz et al. (2009) to real OCO-2 measurements with the goal of improving the precision and accuracy of OCO-2 measurements and thus it is scientifically relevant, as current XCO2 retrieval biases are likely still too large to satisfy the demands of the carbon flux model community. Additionally, the implication that perhaps only a single-band instrument may be needed to make high-quality XCO2 measurements from space has significant implications for potential future GHG missions. The manuscript, although brief, is presented well and I recommend publication in AMT after the authors address a few minor and technical issues.

Specific comments:
- Regarding not using a bias correction, you mention wanting to evaluate the "true" retrieval capability but, as you stated, a bias correction is always employed operationally. Did you look at implementing a bias correction for the one-band retrieval and how it impacted the final σ values relative to the three-band retrieval?
R1-We have checked possible bias corrections and their impact. We looked at the dependency of retrieval difference with respect to different retrieval parameters including surface properties, aerosol parameters and meteorological information. For example, XCO2 retrieval differences show relatively high correlation with retrieved surface albedo as shown in Fig.1. With a linear regression method, we see that the std of difference between TCCON can be reduced by 0.03 ppm after applying bias correction with this parameter. Apart from this, footprint dependent biases should also be subtracted. To conclude, a bias correction is also possible for one-band retrievals which will give similar improvement as for the three-band retrievals.

[Figure]

Figure 1. XCO2 retrieval difference with TCCON as a function of surface albedo slop in one-band retrieval.

- What do the aerosol results (AOD, size parameter, height) look like for the one-band retrieval? Are they similar to the three-band retrieval or does the lack of spectral information at 0.76 and 1.61 um cause the one-band retrieval to behave in interesting ways? In Butz et al. (2009) the size parameter is not retrieved so it would be informative to see the DFS for the three aerosol parameters retrieved in your one-band setup. In the end, it's only the XCO2 that matters but this is an important topic that at least deserves a discussion.
R2- Histogram of DFS for the three aerosol parameters, aerosol size parameter, aerosol optical depth and aerosol layer height in the one-band retrieval are shown in Figure 2. The figure is also included in the paper as Figure 5. As described in the paper, DFS for aerosol parameters are typically below 1.0. The variation

ranges for aerosol size parameters, aerosol optical depth (at the 2.06 um band) and aerosol layer height can vary in a range of [3.5, 4.4], (0.0,0.1] and [-10000, 15000]. Compared to the three-band retrievals, those parameters vary within a smaller range and stay close to the prior values. However, as shown in Table 4 in the paper, fitting those parameters are still very important for XCO2 retrievals to account for aerosol scattering effects.

[Figure]

Figure 2. Histogram of DFS of aerosol (top panel), aerosol size parameter(bottom left), aerosol optical depth(bottom middle) and aerosol layer height(bottom right).

- The only spatial results shown are limited to four fall/winter months in 2014 over EMEA (Fig. 5). However, multiple studies have highlighted temporal patterns in OCO-2 errors (e.g. O'Dell et al., 2018). Did you look at other regions (could you show a global map?) and would it be possible to examine at least one full year of data to ensure that the one-band retrieval has no significant seasonal/regional biases relative to the three-band retrieval? Examining more regions with better coverage could reveal places where the one-band retrieval performs better or worse than the three-band, e.g. snow/ice or tropical forests.

R3-Although we agree it would be useful to do a global comparison based on one year of data, currently we do not have the capability to check one-year's global data between three-band and one-band retrievals since we did not process one year of data because the requirement on calculation resources is quite demanding. On the other hand, TCCON is still considered as the most important tool to evaluate the performance of satellite XCO2 retrievals, and we have used an extensive data set around the different TCCON stations. This already gives a good indication about performance (and difference between 1- and 3-band retrievals) and also seasonal and regional biases. For the last aspect, we check the seasonal relative accuracy (SRA) which is defined by **Dils et al. (2014)** as the standard deviation of biases among four seasons. The SRA value is a good indicator of the variability of the bias in both space and time. For the one-band retrieval in the paper, the SRA is 0.65 ppm which is close to 0.69 ppm found in three-band retrievals before bias correction.

- P3 L33: What percent of soundings are removed by comparing the non-scattering 2.06 um CO2 retrieval to CarbonTracker and filtering the ratio between 0.96 and 1.04? And how were the 0.96 and 1.04 thresholds determined? While this range is several ppm of XCO2, potentially real signals (e.g. large power plants) might be filtered out.

R4-By filtering with the ratio between non-scattering retrieval and CarbonTracker under a range of [0.96, 1.04] about 45% of the converged cases will be removed by applying this filtering option. The 4% difference with CarbonTRacker model is a difference of around 16 ppm. Considering the accuracy of CarbonTRacker of around 3 ppm (see **Peters et al. (2007))** and potential $XCO_2$ variation introduced by large power plants (a few ppm), we think the range used here is still reasonable.

- P3 L34: Could you include a physical explanation of how pre-filtering on CO2 and H2O ratios derived at 2.08 and 2.05 um works?

R5- In one-band cloud filtering, we divide the 2.06 um band into one weak absorption band around 2.08 um and one strong absorption band around 2.05 um. The idea is similar as three-band cloud filtering that a large deviation can be introduced to CO2 and H2O columns retrieved from these two bands with non-scattering retrieval due to different light path sensitivity, see **Taylor et al. (2016)**. A new sentence "The idea is similar as three-band cloud filtering that a large deviation can be introduced to CO2 and H2O columns retrieved from these two bands due to different light path sensitivity." is added in P3 L29 to explain this.

- Regarding Fig. 3, do you have a hypothesis as to why the one-band retrieval does poorly over Lauder and Ascension?

R6- We do not have an explanation for this but it is unlikely due to aerosols since we see similar bias in non-scattering retrievals as well and over ocean aerosols should lead to underestimation instead of overestimation (Butz et al, 2013). We add a sentence "The causes for large biases over the two sites is still unclear but it is unlikely due to aerosols because non-scattering retrievals exhibit similar biases and over ocean aerosols should lead to underestimation instead of overestimation (Butz et al, 2013). " in the paper P4 L32 to mention this as well.

Technical comments:
Overall: define acronyms and technical terms before use. E.g. OCO-2, SD, "full-physics", DFS
P1 L3: change to "A-band", and on P5 L7
P1 L6: change to "ground-based"
P1 L9: remove "region"
P1 L10: Last sentence doesn't make sense
P1 L13: change to "Over the past decade"
P2 L8: change to "simpler"
P4 L7: change to "cloud filter"
P5 L17: change to "has a much"
P5 L22: change to "hundreds of thousands"
P5 L25: "and"

R7-Texts have been adjusted accordingly.

Figure 4: third panel, what do the aerosol layers at approximately -10000 meters represent?
R8-In the retrieval, aerosol layers follow Gaussian layer height distribution under a fixed with of 2000 m. A negative central layer height represents aerosol layers close to the surface.

Figure 5: I would recommend using a perceptually uniform colormap for plotting XCO2 (like you did for the third panel in this figure) and reducing the range so that differences are more visible. Please put a label and units on the colorbars as well.

R9-The figure has been adjusted accordingly.

**Reference**

Dils, B., Buchwitz, M., Reuter, M., Schneising, O., Boesch, H., Parker, R., Guerlet, S., Aben, I., Blumenstock, T., Burrows, J. P., Butz, A., Deutscher, N. M., Frankenberg, C., Hase, F., Hasekamp, O. P., Heymann, J., De Mazière, M., Notholt, J., Sussmann, R., Warneke, T., Griffith, D., Sherlock, V., and Wunch, D.: The Greenhouse Gas Climate Change Initiative (GHG-CCI): comparative validation of GHG-CCI SCIAMACHY/ENVISAT and TANSO-FTS/GOSAT $CO_2$ and $CH_4$ retrieval algorithm products with measurements from the TCCON, Atmos. Meas. Tech., 7, 1723–1744, https://doi.org/10.5194/amt-7-1723-2014, 2014.

Peters, W., Jacobson, A.R., Sweeney, C., Andrews, A.E., Conway, T.J., Masarie, K., Miller, J.B., Bruhwiler, L.M., Pétron, G., Hirsch, A.I. and Worthy, D.E., 2007. An atmospheric perspective on North American carbon dioxide exchange: CarbonTracker. Proceedings of the National Academy of Sciences, 104(48), pp.18925-18930.

Taylor, T. E., O'Dell, C. W., Frankenberg, C., Partain, P. T., Cronk, H. Q., Savtchenko, A., Nelson, R. R., Rosenthal, E. J., Chang, A. Y., Fisher, B., Osterman, G. B., Pollock, R. H., Crisp, D., Eldering, A., and Gunson, M. R.: Orbiting Carbon Observatory-2 (OCO-2) cloud screening algorithms: validation against collocated MODIS and CALIOP data, Atmos. Meas. Tech., 9, 973–989, https://doi.org/10.5194/amt-9-973-2016, 2016.

Butz, A., Guerlet, S., Hasekamp, O. P., Kuze, A., and Suto, H.: Using ocean-glint scattered sunlight as a diagnostic tool for satellite remote sensing of greenhouse gases, Atmos. Meas. Tech., 6, 2509–2520, https://doi.org/10.5194/amt-6-2509-2013, 2013.